# Neurosurgical Clinical Trials for Glioblastoma: Current and Future Directions

**DOI:** 10.3390/brainsci12060787

**Published:** 2022-06-15

**Authors:** Ashish H. Shah, John D. Heiss

**Affiliations:** 1Surgical Neurology Branch, National Institute of Neurological Disorders and Stroke, National Institutes of Health, Bethesda, MD 20892, USA; ashah@med.miami.edu; 2Department of Neurosurgery, University of Miami School of Medicine, Miami, FL 33136, USA

**Keywords:** glioblastoma, brain mapping, temozolomide, radiotherapy, connection-enhanced delivery, immunotherapy, tissue-treating fields, blood–brain barrier opening

## Abstract

The mainstays of glioblastoma treatment, maximal safe resection, radiotherapy preserving neurological function, and temozolomide (TMZ) chemotherapy have not changed for the past 17 years despite significant advances in the understanding of the genetics and molecular biology of glioblastoma. This review highlights the neurosurgical foundation for glioblastoma therapy. Here, we review the neurosurgeon’s role in several new and clinically-approved treatments for glioblastoma. We describe delivery techniques such as blood–brain barrier disruption and convection-enhanced delivery (CED) that may be used to deliver therapeutic agents to tumor tissue in higher concentrations than oral or intravenous delivery. We mention pivotal clinical trials of immunotherapy for glioblastoma and explain their outcomes. Finally, we take a glimpse at ongoing clinical trials and promising translational studies to predict ways that new therapies may improve the prognosis of patients with glioblastoma.

## 1. Introduction

Standard neurosurgery for cerebral glioma requires maximal safe tumor resection. For low-grade tumors (WHO Grade II–III), maximal safe resection of the tumor confers an improved outcome without compromising functional outcomes [1,2]. In the case of glioblastoma, the location of the bulk of the tumor relative to eloquent brain areas dictates the safest and most effective surgical approach. Here, we define eloquent cortex as areas with readily evident neurological function that if damaged results in “disabling neurological deficit” such as corticospinal tracts or dominant receptive/language areas [3]. Jelsma and Bucy reported in 1969 that with glioblastoma (WHO Grade IV glioma), extensive surgery and postoperative radiation extended survival time to 7.5 months compared to 2.5 months with limited resection and postoperative radiation [4]. Survival was 10.5 months in non-central tumors in eloquent brain areas and 4 months with central tumors, those involving the precentral gyrus, Broca’s area, or the Sylvian fissure of the dominant hemisphere. Before surgery, the proportion of patients in excellent or good condition was 31% with non-central and 26% with central tumors. At 3 months after surgery, 78% of patients in the patient group with non-central tumors were in excellent or good condition compared to only 42% of patients with central tumors. Extensive surgery improved function by removing tumors’ mass effect and reducing elevated intracranial pressure.

The role of radiotherapy in glioblastoma therapy was confirmed in 1979 by Walker, Strike, and Sheline. At that time, survival time was 18 weeks without radiation, 28 weeks with 50 Gy, 36 weeks with 55 Gy, and 42 weeks with 60 Gy [5]. Almost three decades later, Stupp et al. verified the survival benefit of more extensive tumor resection results in 2005 in their study of 573 patients with glioblastoma who were randomized to receive radiotherapy (60 Gy) versus radiotherapy (60 Gy) plus temozolomide. Median survival in the temozolomide and radiotherapy arm was 15.8 months in the resection group and 9.4 months in the biopsy-only group. In patients who received radiotherapy (RT) alone, surgical resection (S) improved survival by 12.9 months (S + RT) vs. 7.9 months in the RT alone group [6].

Nevertheless, whole-brain radiotherapy has a greater chance of precipitating cognitive dysfunction from bihemispheric injury than focal brain irradiation. Since tumor recurrences occur most often within 2 cm of the postoperative resection cavity, partial brain irradiation to the volume of 60 Gy in 30 fractions is recommended in patients less than 70, and 40 Gy in 15 fractions for patients over 70 years of age with good performance status [7]. Glioblastoma treatment with maximal safe surgical resection followed by concurrent radiotherapy and temozolomide (TMZ) chemotherapy has been standard for glioblastoma since the 2005 Stupp study. Although this study demonstrated that surgery, radiotherapy, and chemotherapy extended survival in glioblastoma, their effects are limited, and the median life expectancy was 14.6 months [6]. The rest of this review describes the subsequent efforts to improve neurosurgical treatment of glioblastoma further.

## 2. Development and Refinement of Glioblastoma Treatments

### 2.1. Efforts to Improve the Completeness of Tumor Resection

#### 2.1.1. Cortical and Subcortical Electrical Stimulation Mapping

Over the past decade, a consensus agreement was reached among authors in the neurosurgical literature that more extensive tumor resection improves glioblastoma life expectancy. Two separate studies from the UCSF and MD Anderson Cancer Center reported that MRI-visible glioblastoma resection needed to reach a threshold of at least 78%, preferably greater (98 to 100%), to provide a relevant survival benefit (Table 1) [8,9]. (There is abundant evidence supporting maximal safe resection in non-eloquent brain regions. Neurosurgeons have recently advocated for the supramaximal resection of a glioma when feasible to improve overall survival further. Several groups advocate resecting a substantial margin beyond the contrast-enhancing rim for non-eloquent-location high-grade gliomas (~1–2 cm). In a series of reports on this subject, supramaximal resection resulted in patient survival of 20.9–30.7 months [10,11,12]. However, supramaximal resection is impossible in all patients because a bulk tumor invades into the adjacent eloquent cortical and subcortical structures. The survival advantage of the complete resection of MRI-visible tumors in glioblastoma was reported to be 2.9 months in one study and 6.4 months in another [6,8]. The survival advantage of complete tumor resection is much longer in lower-grade gliomas than in glioblastoma (Table 1).

The goal of maximal tumor cytoreduction from the non-eloquent brain depends on accurate knowledge of the relationship between the margin of the eloquent brain and the margin of the tumor. To maximize tumor removal from the non-eloquent brain, the neurosurgeon utilizes techniques to delineate the margin between the tumor and eloquent cerebral cortex and subcortical white matter tracts. Intraoperative electrical stimulation mapping techniques, perfected to identify the eloquent cortex during awake epilepsy surgery procedures, were applied to patients with brain tumors [13]. Electrical stimulation positively affects the motor cortex, evoking movement, and the somatosensory cortex, producing localized paresthesias. Electrical stimulation has negative effects on the completion of language and memory tasks. The margin width between a language site and the resection margin was found to determine the risk of a temporary or permanent language deficit after the resection. No deficits occurred with 20 mm or broader margins, temporary deficits with margins between 20 and 7 mm, and permanent deficits with 7 mm or less [13]. Sanai and colleagues used language mapping during glioma resection and a margin of 1 cm from language areas. Only 1.6% of glioma patients had a persistent language deficit at 6 months after awake tumor resection [8]. The avoidance of brain injury outside the resection cavity depends on the preservation of the arteries of passage and prominent cortical veins (veins of Labbé and Trolard) and recognition of adjacent cortical/subcortical tracts. Modern anesthetic techniques allow patients to be awake for electrical mapping and tumor resection but unaware of the rest of the surgical procedure. Sedation or general anesthesia during the craniotomy’s opening and closing improves patient comfort and cooperativeness and promotes the maximal extent of resection and minimal morbidity [8,14,15]. Some neurosurgeons prefer their patients to stay awake throughout the procedure and provide short-acting opioids like remifentanil for pain uncontrolled by local anesthesia [16]. Electrical stimulation mapping establishes a border that would result in neurological deficit if crossed. Glioblastoma cells always infiltrate throughout the brain and into eloquent brain regions, and sound judgment prevents straying into eloquent brain regions beyond the safe limit of neurosurgical resection [17].

#### 2.1.2. Assessment of the Extent of Tumor Resection in the Intraoperative MRI Suite

Another surgical adjunct to enhance the extent of tumor resection is intraoperative MRI. A randomized trial of patients with a high-grade glioma confirmed that patients with a complete tumor resection had a longer PFS than patients with a residual tumor (median 226 [162–290] vs. 98 days [92–104], *p* = 0.003). This finding highlights the prognostic significance of complete tumor resection. Although a significantly higher proportion of patients in the intraoperative MRI group had a gross total resection (96% vs. 68%, *p* = 0.023), progression-free survival showed only a trend toward significance (*p* = 0.083). In this study, the patients most likely to benefit from intraoperative MRI were the 28% of patients in the iMRI group who would not have received a gross total resection in the microsurgery group. The other 68% percent of patients in either group had a gross total resection or would not be expected to have different outcomes in terms of progression-free survival [18]. Cortical mapping can also be performed in the intraoperative MRI suite for tumors near eloquent regions. After initial tumor resection, MRI scans are performed, and if residual is detected in a surgically accessible area, more tumor is subsequently removed. Therefore, at most, two intraoperative MRI scanning sessions, one after the initial resection and one after the subsequent resection of the residual MRI-visible tumor, are required to confirm tumor resection from non-eloquent regions [19]. Over the last decade, iMRI has become a mainstay of surgical neuro-oncology and has directly impacted onco-functional outcomes for glioma patients [20].

##### Comparison of Craniotomy for Glioblastoma in the Intraoperative MRI (iMRI) Suite versus Neuronavigation in a Conventional Operating Room


Advantage of iMRI suite vs. Standard Neurosurgical operating room (OR)
−Intraoperative evaluation of completeness of resection of tumors is possible in the iMRI suite but not in the conventional OR
Allows recognition and removal of residual tumor during the same procedureMore substantial resections may improve the prognosis
Disadvantages of iMRI suite vs. Standard Neurosurgical OR
−The duration of surgery is increased by intraoperative MRI scan time and transitions from the operating position to the MRI scanner bore
neuronavigation based on preoperative MRI and frameless stereotaxy can reduce this time by providing guidance for centering incisions and locating the tumor, eliminating initial iMRI scans immediately before surgeryreducing iMRI scan to immediately after resection with and without contrast also saves time and allows iMRI’s quality control function of detecting residual tumor
−Head holders are less adjustable; positioning is less flexible−The operating table in the iMRI suite is firmer than the standard OR table−Special safety considerations are required
MR compatible instrumentsMust avoid the receiving coil contacting the body and looping wires




#### 2.1.3. Use of Fluorescent Labeling and Resection of Fluorescent Labeled Tumor Tissue

Another surgical adjunct to increase the volume of malignant glioma resection is oral 5-aminolevulinic acid (5-ALA). 5-ALA penetrates the blood–brain barrier of the MRI-enhancing tumor volume and highlights the extent of the tumor intraoperatively. 5-ALA is a natural precursor molecule in heme synthesis that is selectively converted to fluorescent porphyrins in malignant or highly metabolic tissue. In a randomized controlled multicenter phase III trial by Stummer and colleagues, the rate of gross total resection was 65% in the 5-ALA group compared to 36% in the white-light microscopy alone group [21]. The gross total resection rate of 65% was slightly less than in the microsurgery control group in the intraoperative MRI study of Senft et al. in 2011 [18]. After surgery, temporary neurologic deficits occurred more frequently after 5-ALA use, consistent with more extensive resections, but, longer term, the 5-ALA group had improved progression-free survival at six months (PFS6), better function, and less need for repeat surgical resection [22]. Stummer’s and other studies of 5-ALA led the FDA to approve the drug as an intraoperative optical imaging agent in patients with suspected high-grade glioma in 2017 [23].


*** Randomized clinical trial.**


##### Using 5-Aminolevulinic Acid (5-ALA) to Highlight Glioblastoma


5-ALA is an adjunct measure for identifying high-grade glioma tissue during a craniotomy
5-ALA is taken up by GBM cells and metabolized to protoporphyrin, which accumulates in tumor tissuePeak uptake 6 hours after preoperative oral administrationGlioma visualized using a filter on the operating microscopeViolet light visualizes protoporphyrin IXDoes not identify low-grade gliomaThe brain surrounding the tumor does not enhance with 5-ALA.
Improves removal of the tumor from the non-eloquent brain
Phase III clinical trial of malignant glioma—322 patients *
Microsurgery5-ALA guidedGross total resection36%65%6 m PFS21%41%5-ALA does not affect neurological functionPreserves survival qualityNo difference in serious adverse events and adverse events between the microsurgery and 5-ALA groups



In some centers, other known fluorescent labeling technologies have been proposed such as sodium fluorescein and indocyanine green. Although indocyanine green and fluorescein have lower specificity for gliomas than 5-ALA, both fluorophores may be useful intraoperatively. Sodium fluorescein accumulates in areas of blood–brain barrier disruption, particularly in the contrast-enhancing tumor wall of high-grade gliomas. Fluorescein-guided glioma surgery has been well-described and has been associated with an improved PFS in small retrospective studies [24]. However, because sodium fluorescein is found in the tumor extracellular space, non-specific labelling has been reported after surgical manipulation. The overall sensitivity and specificity of fluorescein for gliomas remains approximately 85% and 90%, respectively [25]. Similarly, indocyanine green has been proposed as an alternative for fluorescent-guided glioma surgery. Although primarily used in cerebrovascular surgery, indocyanine green accumulates within a few minutes after administration due to peritumoral vascular permeability [26]. However, its low half-life, rapid excretion, and non-specific uptake limit its overall applicability to glioma surgery.

#### 2.1.4. Improving Extent of Resection of Gliomas Using Intraoperative Raman Histology

Over the last five years, Raman Histology has been proposed as an important surgical adjunct to improve the extent of the resection of gliomas by identifying tumor infiltration in situ. Raman Histology is capable of rapidly generating histological images of specimens in a label-free manner by detecting molecular vibrations of scattered light. Using this stimulated Raman scattering approach, multicolor images are generated that are comparable to conventional Hematoxylin and Eosin staining [27,28]. As such, serial tumor sampling around the tumor margin is feasible and can permit rapid intraoperative tumor diagnoses [29,30,31]. Similar techniques are also being developed using a hand-held device capable of delineating glioma Raman spectra intraoperatively [32,33]. Overall, these techniques may facilitate the detection of glioma infiltration and, ultimately, improve outcomes for patients by improving the extent of resection.

### 2.2. Efforts to Prevent Neurological Deficits Resulting from Tumor Resection

Protecting quality of life and onco-functional status is critical for patients with malignant gliomas [34,35,36]. The decision to opt for aggressive surgical resection must be counterbalanced by the risks of diminishing the patient’s neuropsychological and functional status. McGirt and colleagues highlighted the effect of a surgically-induced neurologic deficit on survival after surgical treatment of glioblastoma. They retrospectively reviewed 306 consecutive patients, 18 to 70 years of age, with newly diagnosed glioblastoma and good performance documented by Karnofsky performance scores (80–100). Although the 89% of patients who were deficit free after surgery had a 12.8-month median survival, the 5% of patients with a new language deficit had a 9.6-month median survival, and the 6% of patients with a new motor deficit had a 9.0-month median survival. After glioblastoma surgery, a permanent neurological deficit shortened survival by 3 to 4 months and reduced quality of life [37].

Mapping techniques identify eloquent cortex and subcortical tracts involved in expressive and receptive language, motor function, and tactile sensation that are avoided to prevent a neurological deficit. Resections with less than a 1 cm margin from these eloquent cortical areas risk temporary or permanent neurological deficits, with temporary deficits from procedural edema and permanent deficits due to microvascular disruption and resection margin infarcts. Recent research shows that some patients with preserved language, motor, and somatosensory function experience disabling cognitive losses after surgical resections. Cognitive deficits arise from the disruption of cortical networks involved in executive function, attention, default state non-goal oriented tasks, limbic function, salience, and sensorimotor and visual function [38]. The recent integration of MR-based connectomics and diffuse tensor imaging has facilitated the detection of vital subcortical tracts and can help with operative planning [39,40,41]. Operative approaches may be tailored to avoid some of these networks, and judgment is required to assess if the cognitive deficit risk justifies tumor resection interrupting one or more networks.

### 2.3. Less Invasive Glioblastoma Surgical Treatments

#### Laser Interstitial Thermal Therapy

For some deep-seated inoperable glioblastomas, tailored surgical approaches to minimize adjacent white matter disruption while maximizing cytoreduction should be considered. Over the past several years, laser interstitial thermal therapy (LITT) has been popularized for gliomas. Using stereotactic navigation through a 3 mm incision, a laser catheter can be inserted into a target lesion, which can then be coagulated with real-time MR thermography. Although restricted to smaller lesions (<2.4 cm), LITT is particularly suited for treating deep, surgically inaccessible tumors [42,43]. Initial experiences using LITT for gliomas suggest that adequate cytoreduction (>70%) can improve overall survival. Overall survival in newly diagnosed glioblastomas was reported as between 14–24 months in some series. Survival increased more in patients with smaller lesions and there was a greater extent of ablation [44,45,46]. For patients with deep lesions who would otherwise receive a biopsy without tumor resection, LITT can provide cytoreduction that facilitates subsequent chemoradiation. Clinical studies also suggest that LITT may incite or potentiate a local immune response and transiently open the blood–brain barrier to systemic immune cells [44,47,48]. Since the LITT incision is tiny and blood flaps are unnecessary, chemoradiation can be started within 7–10 days of LITT, allowing patients receiving LITT to be treated sooner after the cytoreduction procedure than patients undergoing conventional resections through much larger surgical openings.

### 2.4. Non-Surgical Glioblastoma Treatments

#### 2.4.1. Tumor-Treating Electric Fields

Tumor-treating electric fields disrupt cancer cell division. A randomized trial in GBM patients previously treated with chemoradiotherapy showed that patients treated with the tumor-treating fields (TTFs) and temozolomide (TMZ) had a median progression-free survival of 7.1 months compared to 4.0 months with TMZ alone (*p* = 0.001). Median survival was 20.5 months in the TMZ plus tumor-treating fields and 15.6 months in the TMZ alone group (*p* = 0.004). There was a 43% incidence of mild to moderate skin reactions and a 2% incidence of severe skin reactions (medical device site reactions beneath the transducer arrays) in patients treated with tumor-treating fields plus temozolomide [49]. Seizures and headaches were more frequent in the group treated with tumor-treating fields. In 2011, the FDA approved, under the Premarket Authorization (PMA), a pathway treatment of recurrent or progressive glioblastoma (GBM) using a TTF delivery system. In 2015, the FDA expanded the device’s approval to include the treatment of newly diagnosed GBM when combined with TMZ. This new treatment extends the survival of GBM patients for several months, although skin reactions, prolonged machine use, and other side effects can deter its use.

#### 2.4.2. Immunotherapy and Virotherapy

Immunotherapy using immune checkpoint inhibitors is FDA-approved for treating metastatic melanoma and other cancers. Thus far, clinical trials of immune checkpoint inhibitors in patients with GBM have been unsuccessful. However, there is enthusiasm about developing immunotherapy for GBM because of the limited effectiveness of the current standard therapy of surgical resection of the primary tumor mass and chemoradiation of the residual tumor. Immunotherapy depends on the established capacity of activated lymphocytes to freely enter and exit the central nervous system (CNS) through the blood–brain barrier. Immune checkpoint inhibitors suppress the immune activation of tumors. Checkpoint inhibitors include cytotoxic T-lymphocyte antigen 4 (CTLA-4) and programmed death 1 (PD-1). Ipilimumab, a monoclonal antibody against CTLA-4, received FDA approval in 2011 to treat metastatic melanoma. Nivolumab and pembrolizumab are monoclonal antibodies inhibiting the PD1 receptor that received FDA approval in 2014, and also for the treatment of malignant melanoma. The Phase III trial of nivolumab versus bevacizumab (anti-vascular endothelial growth factor A (anti-VEGF-A) humanized monoclonal antibody) in 369 randomized patients with glioblastoma at first recurrence following standard radiation and temozolomide therapy demonstrated a higher objective response with bevacizumab (23.1%) than with nivolumab (7.8%). The 12-month overall survival (OS) was 42% in both groups [50]. A single immune checkpoint inhibitor such as nivolumab was ineffective. Additional clinical trials using Chimeric Antigen Receptor-T-Cells (CAR-T) have been proposed for glioblastoma. Phase 1 clinical trials targeting the EGFR-VIII mutation have been conducted with modest results (median overall survival 6.9–8 months) [51,52]. Other clinical trials focusing on several epitopes such as GD2, CD147, and B7-H3 are currently being conducted [53]. However, CAR-T cell therapy is restricted by several important factors including the limited penetration of solid tumors, immunosuppressive tumor microenvironment, and heterogenous expression of tumor antigens. Future immunotherapy trials for glioblastoma must employ strategies that enable stronger immune reactions to tumor cells. Immunotherapy may be enhanced by other treatments such as LITT and tumor-treating fields and be more effective as a treatment adjuvant than as a sole treatment. Combination immunotherapeutics that utilize antisense oligonucleotides have also been proposed to target IGF type 1 receptors. The recent IGV-001 trial treated autologous tumor cells with antisense oligonucleotides against IGF-001 ex vivo, irradiated the tumor cells, and reimplanted the cells intraperitoneally using biodiffusion chambers. In the highest dose cohort, overall survival and PFS were 38.2 months and 17.1 months, respectively, for newly diagnosed gliomas [54].

Other efforts to improve outcomes for glioblastoma have relied on viral-based gene therapy and oncolytic virotherapy. Initial studies focusing on viral-based gene therapy have relied on replication-defective adenoviral vectors, which did not demonstrate significant tumor transduction beyond the injection site [55]. However, with the advent of replication-competent viruses, virotherapy may adapt to the evolving tumor microenvironment. Newer generation viral-based gene therapies used replication-competent retroviruses (Maloney murine leukemia virus) and herpes simplex virus to transduce host cancer cells [56,57]. Prodrug activating viral-based gene therapy facilitates tumor selective viral transduction and introduces a “suicide” transgene that converts a non-toxic prodrug into a intracellular chemotherapeutic. The recent Toca511 Phase III clinical trial evaluated the efficacy of a retroviral-mediated gene therapy for recurrent glioblastoma and did not reach its study endpoints [58]. However, there was a significant survival benefit in IDH-mutant and anaplastic astrocytoma. Therefore, selecting the proper patient/subgroup for gene therapy trials remains essential.

Novel oncolytic virotherapies that exploit immunotherapy have recently been described, suggesting that certain virotherapies can induce the adaptative and innate immune response. Of these, oncolytic herpes simplex viruses such as RQNestin have demonstrated increased natural killer cells and tumor-infiltrating macrophages and lymphocytes shortly after viral injection [59,60,61]. Other viral vectors utilizing replication-competent adenoviral vectors (Delta24-RGD) are also being investigated in recurrent gliomas. A recent Phase I clinical trial in current gliomas demonstrated that nearly 20% of patients had tumor responses after treatment with Delta24-RGD with increased peritumoral cytokine levels and tumor-infiltrating lymphocytes [62]. Additionally, other oncolytic virotherapies such as Delta-24-ACT that co-express immunostimulatory ligands are also being investigated in preclinical models to stimulate a robust antitumor immune response [63].

#### 2.4.3. Methods to Improve the Delivery of Therapeutic Agents to Glioblastoma

Clinical trials have tested methods enhancing the delivery of hydrophilic, high molecular weight compounds to brain tumors. These methods include convection-enhanced delivery, blood–brain barrier opening, chemotherapeutic modifications and conjugations that improve the transport of the active antitumor moiety, and osmotic or receptor-mediated opening of the blood–brain barrier [64,65,66,67,68]. Still, the new agents remain less effective than systemic chemotherapy using the hydrophobic agent temozolomide (Table 2).

Intratumoral drug delivery allows the use of drugs that would not be able to penetrate the blood–brain barrier due to their hydrophilic nature or high molecular weight. Convection-enhanced delivery can spread the therapeutic agent more widely throughout the tumor and brain than other methods of intratumoral drug delivery such as intracerebral bolus injection and slow-release polymers. Several modifications of convection-enhanced delivery catheters have been developed recently including a multi-port catheter capable of infusing several reagents simultaneously. A small phase 1 trial using this system demonstrated adequate tumor penetration, minimal back-flow, and a reasonable safety profile [69]. However, intratumoral drug delivery has had limited antitumor effects, either because the agents used were insufficiently tumoricidal or could not be delivered to the entire tumor volume (Table 3).

#### 2.4.4. A Better Understanding of Tumor Components, Therapeutic Susceptibilities, and Mechanisms of Therapeutic Benefit May Lead to Improved Therapeutic Strategies for Glioblastoma

Small (1 cm^3^) and medium (50 cm^3^) glioblastomas follow radial linear, exponential, and Gompertzian growth curves, whereas tumor growth decelerates in large (125 cm^3^) glioblastomas, following a Gompertzian curve. In a recent study, 85% of glioblastomas with central non-contrast-enhancing areas were significantly larger (23.7 cm^3^) than the 15% of tumors (1.2 cm^3^) without central non-enhancing areas. Surprisingly, the non-contrast-enhancing component of the glioblastoma tumor grows at a faster rate than the contrast-enhancing tumor [70].

Cancer is a cellular disease whose cure requires the lethal treatment of every tumor cell. Substantially prolonged survival in glioblastoma depends on preventing tumor recurrence by eradicating tumor cells in the primary tumor mass and the surrounding and distant brain regions. Conventional surgery and chemoradiation of glioblastoma effectively slow the growth of the tumor by eradicating the fastest dividing tumor cells that create the central mass of the tumor. Chemoradiation targets the fastest dividing cells most amenable to DNA damage, which cannot be repaired between rapid cell divisions. These therapies leave slower-dividing tumor clones to maintain glioblastoma growth. If this theory is correct, the present glioblastoma treatment essentially lengthens survival by eradicating the most rapidly dividing tumor clones. Life expectancy increases after the first wave of therapy because the glioblastoma growth rate falls when slower-dividing tumor clones drive it. If a tumor cure is presently unattainable and radio- and chemotherapy extend life by eliminating the fastest-growing tumor cell clones, therapies that slow the tumor cell cycle through non-DNA toxic treatments may be logical choices for treating recurrent glioblastoma. Future therapies may slow tumor growth by changing the tumor environment, providing time and a more conducive milieu for treatments such as immunotherapy to eradicate glioblastoma.

## 3. Conclusions

The improvement in glioblastoma prognosis and the development of effective therapies have not kept up with the tremendous advances in understanding the genetics, biology, and pathology of glioblastoma. Surgical treatment advances are limited because surgical resection cannot breach eloquent functional cortex and white matter tracts, preventing curative resections. Potentially curative radiotherapy doses are unsafe, causing radionecrosis of critical brain structures involved in the brain’s cognitive and other essential functions. Chemotherapy is often initially partially effective, but glioblastoma becomes progressively resistant to it. As a result, new and better treatments are sorely needed. Ongoing research into the mechanisms of the proliferation, invasion, and immune evasion of glioblastoma will identify new targets for glioblastoma therapy. Proven treatments, unique surgical tools, and novel molecularly-guided treatments provide a foundation for neurosurgeons to build more effective strategies for improving the care and extending the survival of patients with glioblastoma (Figure 1).

## Figures and Tables

**Figure 1 brainsci-12-00787-f001:**
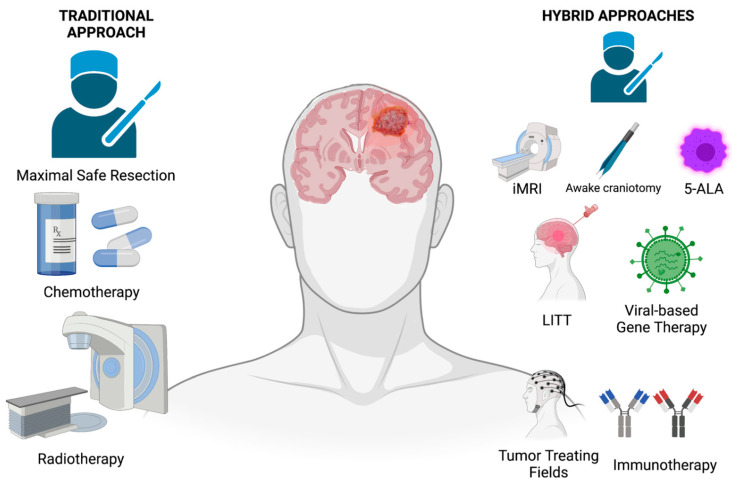
Schematic demonstrating new hybrid approaches for glioblastoma therapy.

**Table 1 brainsci-12-00787-t001:** Prognostic impact of extent of resection.

Brain Tumor Type and WHO Grade	Invasive	Complete Resection Possible	Life Expectancy (Months) Biopsy	Life Expectancy (Months) MR Incomplete Resection	Life Expectancy (Months) MR Complete Resection	Survival Advantage (Months) with MR Complete Resection Compared to Incomplete Resection
I Neuronal DNET Ganglioglioma Pilocytic astrocytoma	No	Yes; if outside eloquent structures	Prolonged	Prolonged	Prolonged	Uncertain: residual tumors require additional surgery
II Low-grade astrocytoma and oligodendroglioma	Yes	No		61	90.5	29.5
III Anaplastic astrocytoma and oligodendroglioma	Yes	No		64.9	75.2	10.3
IV Glioblastoma multiforme	Yes	No		11.3	14.2	2.9
			9.4 ^†^		15.8 ^†^	6.4 ^†^

^†^ [6].

**Table 2 brainsci-12-00787-t002:** Comparison of methods to deliver therapeutic agents to glioblastoma.

	Convection-Enhanced Delivery	BBB Opening	Systemic Chemotherapy
Drug delivery into brain tissue or lesion	During tissue infusion	During the opening of the BBB	Limited by the intact BBB
MW of therapeutic agent	Large or small	Large or small	Small
Brain–Blood Concentration	>100 × systemic concentration	≤1 × systemic concentration	<1 × systemic concentration
Hydrophilic compounds	Enters CNS	Enters CNS	<<<1 × systemic concentration
Hydrophobic compounds	Enters CNS	Enters CNS	<1 × systemic concentration
Distribution of Compound within CNS	Volume spreads radially from the infusion site	The volume of distribution rests in the arterial distributions injected with mannitol	Entire CNS
The volume of the brain that can be treated	Large (4–8 cm^3^)	Large (4–8 cm^3^)	Large (entire brain)

**Table 3 brainsci-12-00787-t003:** Comparison of intracerebral drug delivery techniques.

	Convection-Enhanced Delivery	Bolus Intralesional Therapy	Slow-Release Polymer
Speed of drug delivery into brain tissue or lesion	Hours to days	Seconds	Days to weeks
Means of the spread of drug	Drug moves by bulk flow through the interstitial space	Drug moves by diffusion along concentration gradients	Drug moves by diffusion along concentration gradients
Spread by MW	Small = Large MW	Small > Large MW	Small > Large MW
Variability in drug concentration	Homogeneous drug levels (1–100% of infused) within a brain volume	High concentration at infusion point with a steep fall-off in concentration throughout the surrounding brain	High concentration around polymer with a steep fall-off in concentration throughout the surrounding brain
Depth of penetration of drug	15–20 mm	1–4 mm	1–4 mm
The volume of the brain that can be treated	Large (4–8 cm^3^)	Small (mm^3^)	Small (mm^3^)

## Data Availability

Not applicable.

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
