# Peer review of "Neurosurgical Clinical Trials for Glioblastoma: Current and Future Directions"

_brainsci, 2022, doi:10.3390/brainsci12060787_

Round 1
Reviewer 1 Report
Thank you for the opportunity to review this manuscript.
It is a thorough review. Here, the authors provide an overview of past, present, and future treatment modalities in glioblastoma.
Although the authors mention 5-ALA in the surgical armamentarium guiding resection through fluorescence guidance, the use of fluorescein as adjuvant fluorophore is omitted. Maybe the authors can consider adding the increasing evidence of fluorescein to this comprehensive review.
Layout and format: The manuscript is structured and meets the expected format of the targeted journal.
Title: The title of the manuscript reflects the content of the article.
Abstract: The abstract is well-structured and reflects the content of the article.
Introduction: The introduction describes the aim of the study accurately.
Methods and statistics: The authors describe data acquisition and the experimental design. The used statistical tests are appropriate and sufficient.
Results: The presentation of the results is clear and stringent. The significant limitations are listed.
My recommendation: Accept after minor revisions.
Author Response
Reviewer #1:
It is a thorough review. Here, the authors provide an overview of past, present, and future treatment modalities in glioblastoma.
Although the authors mention 5-ALA in the surgical armamentarium guiding resection through fluorescence guidance, the use of fluorescein as adjuvant fluorophore is omitted. Maybe the authors can consider adding the increasing evidence of fluorescein to this comprehensive review.
Response: Thank you for this point! We have now added a paragraph regarding the use of fluorescein as an adjuvant fluorophore to differentiate between tumor and normal brain. We have also mentioned limitations of certain fluorophores such as ICG in glioma surgery.
Reviewer 2 Report
The author writes a narrative review for glioblastomas and future trials in the same. There are several shortcomings which undermine the paper
1. The author writes "This approach can completely excise grade I tumors located in the non-eloquent brain. On the other hand, Grade II, III, and IV tumors are infiltrative and cannot be fully resected due to microscopic finger-like projections into adjacent brain parenchyma and solitary tumor cells through the cerebrum. "
The statement is factually wrong and has no reference whatsoever. Statements like this have to be removed and cannot be part of published literature. Daffau's series and many others have shown complete excision of Grade 3 and below lesions with long-term survival even without chemotherapy. Grade 4 may be applicable. Kindly avoid making such sweeping statements that are not founded on facts and maybe just based on self-experience.
2. Most of the paper is just a repeat of previous trials and therapies like 5 ALA, Intraoperative MRI and Tumor treating fields, etc. There is nothing new or even novel in the paper. I don't see any use or addition to literature or even collation of previous data. Even previous published articles by the same publisher are not cited eg: "Deora H, Ferini G, Garg K, Narayanan MDK, Umana GE. Evaluating the Impact of Intraoperative MRI in Neuro-Oncology by Scientometric Analysis. Life (Basel). 2022 Jan 25;12(2):175. doi: 10.3390/life12020175. PMID: 35207463; PMCID: PMC8877236."
Overall it seems like a summary of known facts and literature.
Author Response
Reviewer #2:
1. The author writes "This approach can completely excise grade I tumors located in the non-eloquent brain. On the other hand, Grade II, III, and IV tumors are infiltrative and cannot be fully resected due to microscopic finger-like projections into adjacent brain parenchyma and solitary tumor cells through the cerebrum. "
The statement is factually wrong and has no reference whatsoever. Statements like this have to be removed and cannot be part of published literature. Daffau's series and many others have shown complete excision of Grade 3 and below lesions with long-term survival even without chemotherapy. Grade 4 may be applicable. Kindly avoid making such sweeping statements that are not founded on facts and maybe just based on self-experience.
Response: Thank you. We have amended that statement accordingly and have referenced the Duffau literature in the manuscript. We do agree that a greater extent of resection leads to a significantly longer overall survival. Our point is rather that microscopic tumor infiltration is often unrecognizable intraoperatively; therefore, surgical adjuvant therapies remain necessary especially in the setting of tumor recurrence for lower grade (WHO II/III) lesions.
2. Most of the paper is just a repeat of previous trials and therapies like 5 ALA, Intraoperative MRI and Tumor treating fields, etc. There is nothing new or even novel in the paper. I don't see any use or addition to literature or even collation of previous data. Even previous published articles by the same publisher are not cited eg: "Deora H, Ferini G, Garg K, Narayanan MDK, Umana GE. Evaluating the Impact of Intraoperative MRI in Neuro-Oncology by Scientometric Analysis. Life (Basel). 2022 Jan 25;12(2):175. doi: 10.3390/life12020175. PMID: 35207463; PMCID: PMC8877236."
Response: Thank you for this point. Since this article is a review paper, we focused on historical perspectives that have helped transform modern clinical trials in surgical neuro-oncology. In line with contemporary clinical trials in neurosurgery, we have added newer results of novel biological therapies for brain tumors including oncolytic virotherapies, anti-sense oligonucleotide therapy and immunotherapy. We have also now added several references including the ones mentioned Deora et al. to the manuscript. We have expanded our review to include innovative techniques in surgical neuro-oncology including novel oncolytic virotherapies, stimulated Raman histology and RNAi based technologies.